# Herpes Simplex Virus Type-2 Paralyzes the Function of Monocyte-Derived Dendritic Cells

**DOI:** 10.3390/v12010112

**Published:** 2020-01-16

**Authors:** Linda Grosche, Petra Mühl-Zürbes, Barbara Ciblis, Adalbert Krawczyk, Christine Kuhnt, Lisa Kamm, Alexander Steinkasserer, Christiane Silke Heilingloh

**Affiliations:** 1Department of Immune Modulation, Universitätsklinikum Erlangen, Friedrich-Alexander-Universität Erlangen-Nürnberg, D-91052 Erlangen, Germany; 2Department of Infectious Diseases, University Hospital Essen, University of Duisburg-Essen, D-45147 Essen, Germany; 3Institute for Virology, University Hospital Essen, University of Duisburg-Essen, D-45147 Essen, Germany

**Keywords:** HSV-1, HSV-2, dendritic cells, migration, adhesion, LFA-1, cytohesin-1, CYTIP

## Abstract

Herpes simplex viruses not only infect a variety of different cell types, including dendritic cells (DCs), but also modulate important cellular functions in benefit of the virus. Given the relevance of directed immune cell migration during the initiation of potent antiviral immune responses, interference with DC migration constitutes a sophisticated strategy to hamper antiviral immunity. Notably, recent reports revealed that HSV-1 significantly inhibits DC migration in vitro. Thus, we aimed to investigate whether HSV-2 also modulates distinct hallmarks of DC biology. Here, we demonstrate that HSV-2 negatively interferes with chemokine-dependent in vitro migration capacity of mature DCs (mDCs). Interestingly, rather than mediating the reduction of the cognate chemokine receptor expression early during infection, HSV-2 rapidly induces β2 integrin (LFA-1)-mediated mDC adhesion and thereby blocks mDC migration. Mechanistically, HSV-2 triggers the proteasomal degradation of the negative regulator of β2 integrin activity, CYTIP, which causes the constitutive activation of LFA-1 and thus mDC adhesion. In conclusion, our data extend and strengthen recent findings reporting the reduction of mDC migration in the context of a herpesviral infection. We thus hypothesize that hampering antigen delivery to secondary lymphoid organs by inhibition of mDC migration is an evolutionary conserved strategy among distinct members of Herpesviridae.

## 1. Introduction

Dendritic cells (DCs) are a group of leukocytes operating at the interface of innate and adaptive immunity, possessing an outstanding efficacy to antigen-specifically activate lymphocytes [1,2,3,4]. Under steady state conditions, immature DCs (iDCs) reside in virtually all tissues searching for (non-host) antigens [5]. Maturation of DCs is induced upon antigen uptake or other stimuli, such as engagement of pathogen recognition receptors on/in DCs or the recognition of “danger” signals as well as inflammatory cytokines secreted from infected bystander cells [6]. Mature DCs (mDCs) show increased surface expression levels of MHC class I and II molecules [7,8] as well as the co-stimulatory molecules CD40, CD80, and CD86, which are important for T cell activation [5]. In addition, the surface glycoprotein CD83, which is one of the best maturation markers for mDCs and induces potent immune-modulatory effects, is also strongly upregulated on mDCs [9,10,11,12,13]. During maturation, DCs not only switch from a processing to a presenting state, but also loosen their adhesion to shift to a migratory phenotype [5]. This is accompanied by alterations in their chemokine receptor expression. While inflammatory chemokine receptors, such as C–C chemokine receptors (CCR)1, CCR2, CCR5, CCR6, and C–X–C chemokine receptor (CXCR)1, are downregulated on mDCs, expression of, e.g., CCR7 is strongly induced [14,15]. CCR7 responds to the C–C motif chemokine 21 (CCL21) as well as CCL19 and thus chemotactically directs mDCs toward lymphatic vessels and lymph nodes [16,17,18,19,20].

Leukocyte migration is a prerequisite to successfully combat an infection via immediate initiation of innate and adaptive immune responses [21]. In contrast to the adhesion-dependent two-dimensional leukocyte migration, three-dimensional “amoeboid” migration of leukocytes, and especially mDCs, is different from other cell types, since it is not dependent on adhesive contacts mediated by integrins in porous environments [22,23,24,25,26,27,28,29]. This ability, to switch between integrin-dependent and -independent migration, enables DCs to directly and rapidly follow chemokine gradients, independently of preformed tracks of specific integrin ligands [23,30]. Considering that excessive adhesion in two- as well as three-dimensional environments would immobilize DCs, integrin-mediated adhesion has to be tightly controlled [31,32].

Integrins are heterodimeric transmembrane proteins that are ubiquitously expressed on almost all cells in Metazoa [33]. Leukocytes express integrins of the β1, β2, β3, and β7 families, while β7 and β2 integrins are unique to leukocytes [34,35]. The latter one is the predominant integrin subfamily expressed on leukocytes and thus possesses essential immune functions [36,37]. The β2 integrins are composed of the β subunit CD18 [38] that associates with one of four different α chains (CD11) to form LFA-1 (CD11a (ITGAL)/CD18 or αLβ2), Mac-1 (CD11b (ITGAM)/CD18 or αMβ2), p150.95 (CD11c (ITGAX)/CD18 or αXβ2), and αDβ2 (CD11d (ITGAD)/CD18). Members of β2 integrins bind to the intercellular adhesion molecules-1–5 (ICAM-1–5), which are, amongst others, expressed by endothelial cells and leukocytes. Additionally, β2 integrins are also able to recognize other proteins, such as fibronectin and polysaccharides [36,39,40]. A complete absence, or at least a significant reduction, of β2 integrin expression on leukocytes in humans is associated with the leukocyte adhesion deficiency I (LAD I). Patients show severe recurrent infections and a defect in tissue repair. This reveals the importance of β2 integrins in immune surveillance [41,42,43,44].

Like almost all other integrins, β2 integrins switch between their active and inactive conformation [45,46]. During outside-in signaling, integrin engagement with its specific ligand triggers complex intracellular signaling pathways and induces the final stage of integrin activation [47,48]. Since integrins lack any known intrinsic enzymatic activity, a variety of signaling molecules is recruited to the cytoplasmic tail to elicit distinct downstream events [49,50,51]. Thus, integrins are able to integrate the signals received from the cellular microenvironment into intracellular responses, initiating, e.g., cell migration, survival, or proliferation [47,52]. Moreover, the process designated inside-out-signaling modifies the activation state of the integrin, i.e., affinity and avidity, for ligand binding [50,51]. Two proteins expressed by mDCs are well described to contrary control β2 integrin activity via inside-out-signaling, namely cytohesin-1 and cytohesin-1 interacting protein (CYTIP) [53,54,55,56,57]. Binding of cytohesin-1 to the intracellular tail of CD18 promotes ligand binding of β2 integrins, whereas CYTIP reverses this activation by binding cytohesin-1 to recruit it to the cytosol. Via siRNA knock-down experiments, CYTIP was indeed proven to be essential for inactivation of β2 integrin activity as well as controlling adhesion [57,58].

Herpes simplex virus (HSV) constitutes the prototypic genus of the human pathogenic α-herpesvirinae subfamily. The two closely related viruses HSV type 1 (HSV-1) and type 2 (HSV-2) are widespread throughout the population, with seroprevalences ranging from 10% to 90%, depending on gender, age, and geographic region [59,60,61,62,63,64,65]. Primary HSV-1 infection generally occurs during early childhood and is usually an asymptomatic inapparent orolabial infection, via direct contact of infectious lesions/body fluids [66,67]. HSV-2 infections are often associated with transient genital lesions, since HSV-2 is among the most frequent sexually transmitted infections, and often linked with a higher risk for HIV acquisition [68,69]. However, during the last years it has become clear that the classification between HSV-1-caused orofacial herpes versus HSV-2-caused genital herpes is not as strict as supposed, since both viruses are able to induce similar symptoms [70,71]. Initial as well as recurrent HSV-1/-2 infections can either be subclinical or associated with mild symptoms manifesting as herpetic lesions, whereas immuno-compromised/-immature individuals can suffer from life-threatening neurological illnesses [66,71,72,73]. Upon access to the host cell, HSV-1 and HSV-2 replication is characterized by a fast tripartite gene expression cycle during productive infection. Accordingly, gene products of the immediate early (IE) phase are essential for an entry into the early (E) phase, which in turn give rise to late (L) gene expression [74]. Apart from productive infection, HSV-1 and HSV-2 are neurotropic herpesviruses with the ability to establish life-long latency within neurons in sensory ganglia, via retrograde viral spread for neuroinvasion. Periodic reactivation induces viral replication and anterograde spread to mucocutaneous sites for reinfection of epithelial cells [62,75,76].

Considering their pivotal role in the orchestration of immune responses, DCs constitute promising targets for immune-evasion strategies evolved by herpesviruses. Amongst others, HSV-1 inhibits DC maturation, mediates the loss of CD83, and modulates two additional hallmarks of mDC biology, i.e., adhesion and migration. The constitutive activation of the β2 integrin LFA-1 (lymphocyte function-associated antigen 1), via the reduction of the negative regulator CYTIP upon HSV-1 infection, was shown to account for an increased adhesiveness and thereby a reduced migratory capability of infected mDCs [57]. Moreover, also the human cytomegalovirus (HCMV), a β-herpesvirus, triggers similar modulations of DC function and biology to delay the initiation of an antiviral immune response [77,78,79,80].

In the present study, we extend our recent knowledge of herpesviral-mediated modulations of DC biology. We demonstrate for the first time that also HSV-2 mediates a proteasomal degradation of CYTIP in mDCs very early upon infection. Due to this, HSV-2-infected mDCs reveal significantly induced activation of β2 integrins. Via this manipulation of the ligand binding state of β2 integrins, HSV-2 is able to induce mDC adhesion on specific ligands, i.e., fibronectin or ICAM-1. Hence, this leads to a reduced migration capability of infected mDCs very early after infection. At later time points during infection, HSV-2 further downregulates chemokine receptor expression of CCR7, recognizing the lymphoid tissue expressed chemokine CCL19. Thus, HSV-2 paralyzes normal DC functions to render these cells unable to migrate toward areas with high lymphocyte density, which is essential for the induction of an efficient antiviral immune response.

## 2. Materials and Methods 

### 2.1. Generation of Monocyte-Derived Dendritic Cells (DCs)

As reported earlier, DCs were generated from peripheral blood mononuclear cells (PBMCs) from leukoreduction system chambers (LRSCs) of healthy donors, using Lymphoprep (Nycomed Pharma, Zurich, Switzerland) for density centrifugation [81]. Briefly, PBMCs were allowed to adhere on standard tissue-flasks (Nunc, Thermo Fisher Scientific, Langenselbold, Germany) while culturing in DC-medium (RPMI 1640 medium (BioWhittaker, Lonza, Verviers, Belgium) supplemented with 100 U/mL penicillin, 100 mg/mL streptomycin (Sigma-Aldrich, Steinheim, Germany), 1% (*v*/*v*) AB serum (Sigma-Aldrich), 2 mM L-glutamine (Lonza), and 10 mM HEPES pH 7.5 (Sigma-Aldrich)) for 1 h. Subsequently, the non-adherent cell fraction was rinsed off with RPMI 1640 and adherent monocytes were differentiated into DCs as described elsewhere [82,83]. Briefly, immature DCs (iDCs) were harvested 4 days post adherence. Mature DCs (mDCs) were generated by adding 40 U/mL GM-CSF, 250 U/mL IL-4, 10 ng/mL TNF-α (Peprotech, Hamburg, Germany), 1 µg/mL prostin E2 (Pfizer, Berlin, Germany), 200 U/mL IL-1β (Cell-Genix, Freiburg, Germany), and 1000 U/mL IL-6 (Cell-Genix) to the medium, and mDCs were used 2 days post induction of maturation (6 days post adherence).

### 2.2. Viruses

The HSV-2 strain G and the HSV-1 strain 17+CMV-GFP/UL43 (BioVex) [84,85], containing a CMV promoter-driven GFP reporter cassette inserted into the UL43 locus of HSV-1 strain 17+, were used in this study. The latter allows the evaluation of the infection rate based on the percentage of directly infected GFP-positive cells. Virus stock preparation and titer determination were performed as recently described [86]. For ultraviolet (UV) inactivation of the virions (HSV-1 UV, HSV-2 UV) HSV-containing suspensions were irradiated with defined doses of UV light (1.200 J/cm^2^; 8 × 0.12 J/cm^2^). 

### 2.3. HSV-1 and HSV-2 Infection of iDCs and mDCs

Infection experiments of iDCs or mDCs were performed as recently described [82,83]. Briefly, iDCs or mDCs were harvested and washed in RPMI 1640. Cells (1 × 10^6^–2 × 10^6^) were resuspended in 300–350 µL infection medium (RPMI 1640 supplemented with 20 mM HEPES (N-2-hydroxyethylpiperazine-N-2-ethanesulfonic acid; Lonza)) containing a defined amount of HSV-1 or HSV-2 plaque forming units (pfu). For HSV-1 infection, a multiplicity of infection (MOI) of 1 or 2 was used (as indicated), whereas HSV-2 infection experiments were performed using an MOI of 5. As a mock control, cells were treated with infection medium supplemented with the respective amount of MNT buffer (30 mM MES, 100 mM NaCl, and 20 mM Tris). After 1 h of infection at 37 °C and shaking at 300 rpm cells were collected via centrifugation. Subsequently, cells were cultured in DC medium supplemented with 40 U/mL GM-CSF and 250 U/mL IL-4. To block proteasomal degradation either 10 µM MG-132 (Enzo Life Science, Lörrach, Germany) or 2 µM bortezomib (BZ; Santa Cruz Biotechnology, Heidelberg, Germany) were added 1 hpi (hour post infection). For blocking the ubiquitin E1-activating enzyme, 80 µM PYR-41 (Sigma-Aldrich) was added 4 hpi. As solvent control, dimethyl sulfoxide (DMSO; Sigma-Aldrich) was added to the medium.

### 2.4. Construction of Expression Plasmids

The plasmid constructs pCDNA3.1-CYTIP, containing human full-length CYTIP, and pCDNA3.1-cytohesin-1, containing human full-length cytohesin-1, were prepared as follows: mRNA was isolated from mock-treated mDCs using (RNeasy MiniKit, Qiagen, Hilden, Germany) and subsequently transcribed into cDNA using the First Strand cDNA synthesis kit (Thermo Fisher Scientific). The coding sequences of CYTIP and cytohesin-1 were amplified using sequence-specific primer pairs possessing the respective recognition sites for the indicated restriction enzymes (*EcoRI*-CYTIP-*XhoI*: fw 5′-*GAATTC*ATGTCTTTACAAAGGCTCCTG-3‘, rv 5‘-*CTCGAG*TCAAAAGCGACTTTCTTCCTC-3‘; *HindIII*-cytohesin-1-*EcoRI*: fw 5′- *AAGCTT*ATGGAGGAGGACGACAGCTAC-3‘, rv 5‘-*GAATTC*TCAGTGTCGCTTCGTGGAGGA-3‘). After extraction of the correct PCR products (QIAquick Gel Extraction Kit, Qiagen) from agarose gels, the fragments were ligated into the pCR-Blunt II TOPO vector (Zero Blunt TOPO, Invitrogen, Thermo Fisher Scientific) following the manufacturer’s recommendations. The presence and correct sequence of the inserts were verified via analytic digestion and Sanger sequencing (Eurofins MWG Operon), respectively. Correct inserts encoding either CYTIP or cytohesin-1 were ligated into the pcDNA3.1 expression vector (Invitrogen), harboring the CMV enhancer–promoter, by using the Rapid DNA Ligation Kit according to the manufacturer’s instructions (Roche, Mannheim, Germany). 

### 2.5. Transient Transfection and HSV-1 or HSV-2 Infection of HEK293T Cells 

For transfection, HEK293T cells were seeded into 6-well plates and cultured in Dulbecco’s minimal essential medium (Lonza) supplemented with 10% FBS (FBS Superior Biochrom, Merck, Darmstadt, Germany), 2 mM L-glutamine, 100 U/mL penicillin, and 100 mg/mL streptomycin (D10 medium). The next day, HEK293T cells were transfected with 1 µg of pcDNA3.1 plasmid encoding CYTIP or cytohesin-1 using the jetPRIME transfection reagent (Polyplus, VWR International GmbH, Darmstadt, Germany) according to the manufacturer’s recommendations. After 24 h, cells were mock-treated or infected with HSV-1 (MOI of 2), HSV-1 UV (MOI of 2; 8 × 0.12 J/cm^2^), HSV-2 (MOI of 5), or HSV-2 UV (MOI of 5; 8 × 0.12 J/cm^2^). For this, cells were gently washed with 1 mL PBS and 500 µL of the virus-containing inoculum was added per well. After 1 h of infection at 37 °C, the inoculum was aspirated and 2 mL of D10 medium was added per well. To block proteasomal degradation or the ubiquitin E1-activating enzyme, the respective inhibitors were added as described for “HSV-1 and HSV-2 infection of iDCs and mDCs”. 

### 2.6. Mg/EGTA Treatment of mDCs

To induce β2 integrin (LFA-1) integrin activity, mDCs were treated with Mg/EGTA as described earlier [80,87]. In brief, mDCs were incubated in RPMI 1640 supplemented with 20 mM HEPES, 5 mM MgCl2, and 1 mM ethylene glycol-bis(β-aminoethyl ether)-*N,N,N*’,*N*’-tetraacetic acid (EGTA) at 37 °C for 30 min. Afterwards, cells were directly subjected to flow cytometric analyses or fibronectin adhesion assays. 

### 2.7. Flow Cytometric Analyses and Antibodies Used

The purity and maturation status of DCs was analyzed using flow cytometry. To determine the expression levels of specific surface proteins during an HSV infection, cells were harvested at the indicated time points and washed once with ice-cold PBS. Subsequently, cells were stained at 4 °C in the dark for 30 min using the following monoclonal fluorophore-labeled antibodies: CD80-PE/Cy5 (clone L307.4, BD Biosciences, Heidelberg, Germany), CD83-APC (clone HB15e, Invitrogen), CD86-PE (Clone 2331, FUN-1, BD Biosciences), MHCII-APC/Cy7 (clone L243, BioLegend, Fell, Germany), CCR7-PE/Cy7 (clone 3D12, BD Biosciences), CXCR4-APC (clone 12G5, BioLegend), CD11a-PE (clone TS2/4, BioLegend), CD11b-APC-Cy7 (clone ICRF44, BD Biosciences), CD11c-PeCy5 (clone B-ly6, BD Biosciences), or CD18-APC (clone CBR LFA-1/2, BioLegend). For analysis of β2 integrin activity the anti-activated CD11/CD18 clone mAB24 (Hycult Biotech, Beutelsbach, Germany) or the respective isotype control mouse IgG1 κ (clone P3.6.2.8.1, eBiosciences) was used with subsequent incubation with an AlexaFluor647-labelled secondary antibody (Invitrogen). Indicated antibodies were resuspended in ice-cold PBS supplemented with 2% (*v*/*v*) FBS. To discriminate between living versus dead cells, the cells were stained with the LIVE/DEAD Fixable Violet dead cell stain kit (Life Technologies, Darmstadt, Germany). Afterwards, cells were washed twice in ice-cold PBS supplemented with 2% (*v*/*v*) FBS and fixed in ice-cold PBS supplemented with 2% (*w*/*v*) paraformaldehyde (PFA; Alfa Aeasar). Samples were measured using a BD FACS Canto II (BD Biosciences) and data were analyzed using FCS express 5. 

### 2.8. WesternBlot Analyses and Antibodies Used

Cells were harvested at the indicated time points and washed once with ice-cold PBS. Protein lysates were prepared by resuspending the cell pellets in ice-cold NP-40 lysis buffer (10% [*v*/*v*] glycerol; 2 mM EDTA, pH 8; 137 mM NaCl; 50 mM Tris pH 7.2; 0.5% [*v*/*v*] NP-40) freshly supplemented with 20 mM sodium fluoride, 2 mM sodium orthovanadate, and 2 mM phenylmethanesulfonyl fluoride. After a 20-min incubation on ice, protein lysates were cleared by centrifugation at 15,900× *g* at 4 °C for 20 min. Subsequently, the protein amount was determined via a Bradford assay (RotiQuant; Carl Roth, Karlsruhe, Germany). Lysates were adjusted with Aqua bidest. and 4x Laemmli (125 mM Tris-HCl pH 6.8, 4% SDS, 20% [*v*/*v*] glycerin, bromphenol blue, 10% [*v*/*v*] 2-mercaptoethanol) to reach a final protein concentration of 1 µg/µL. Samples were denatured at 95 °C for 10 min. Proteins were separated by SDS-PAGE and transferred to nitrocellulose membranes (GE Healthcare, Solingen, Germany) via Western blotting. For blocking unspecific antibody binding sites, membranes were incubated in 1x RotiBlock (CarlRoth) on a gently rocking platform for at least 1 h. Subsequently, membranes were incubated with the following primary antibodies in 1x RotiBlock at 4 °C and gently rocking overnight: rat hybridoma anti-CYTIP clone 2F9 and rat hybridoma anti-cytohesin-1 clone 7H2 (Helmholtz Center Munich, Munich, Germany), anti-CD83 (clone F-5, Santa Cruz Biotechnology) anti-ICP0 (clone 11060, Santa Cruz Biotechnology), anti-ICP4 (clone 10F1, Santa Cruz Biotechnology), anti-ICP27 (clone H1113, Santa Cruz Biotechnology), anti-ICP8 (clone 10A3, Santa Cruz Biotechnology), anti-gB (clone 10B7, Santa Cruz Biotechnology), and anti-GAPDH (clone 6C5, Millipore, Schwalbach, Germany). Membranes were washed five times each five minutes with 1x TBS-T (20 mM Tris, 150 mM NaCl adjusted to pH 7.4 supplemented with 0.1% Tween-20) for subsequent incubation with the appropriate secondary HRP-labeled antibody (Cell Signaling Technology, Frankfurt am Main, Germany) in 1× TBS-T for 1 h. After washing the membranes as described, signals were detected using Amersham ECL Prime Western Blotting Detection Reagent (GE Healthcare).

### 2.9. Fibronectin Adhesion Assay

Maxisorb 96-well plates (Nunc) were coated with 20 µg/mL fibronectin (Sigma-Aldrich) in PBS at 4 °C overnight. The next day, wells were washed once with PBS and blocked with 0.01% (*w*/*v*) gelatin (Sigma-Aldrich) in PBS for 2 h. Wells were washed once with PBS before cells were allowed to adhere. Mock- or HSV-infected mDCs were harvested 4 and 24 hpi, washed once with RPMI 1640 and adjusted to a cell concentration of 4 × 10^5^ cells/mL in RPMI 1640. Per well, 4 × 10^4^ mDCs/100 µL were allowed to adhere at 37 °C for 45 min. Afterwards, the non-adherent fraction was gently washed off with pre-warmed RPMI 1640. Each experimental condition was performed in quadruplicates. The adherent fraction was quantified using a β-glucuronidase assay as described previously [88]. In brief, adherent cells were resuspended in 25 μL PBS and 5 μL 1% TritonX-100 in PBS. After 10 min, lysates were cleared by centrifugation at 18,400× *g* for 1 min. Subsequently, 25 µL of the lysates were incubated with 75 μL 10 mM 4-nitrophenyl-β-D-glucuronide (NPDG, solved in 100 mM sodium acetate, pH 4; Sigma-Aldrich) at 37 °C for 6 h. After adding 100 μL 0.4 M glycine (pH 10), the absorbance at 405 nm was measured using a Victor^2^ multilabel counter (PerkinElmer, Jugesheim, Germany). As input condition, 100 µL of the mDC suspension was preserved prior to adhesion. A standard curve with defined cell numbers was prepared for each sample using cells of the input condition to calculate the number of adherent cells.

### 2.10. ICAM Adhesion Assay

Maxisorb 96-well plates (Nunc) were coated with 100 µL of 4 µg/mL goat anti-human IgG (Fc specific; Jackson ImmunoResearch Laboratories, Ely, Cambridgeshire, UK) in PBS at 4 °C overnight. After blocking the wells with 100 µL of 1% (*w*/*v*) BSA in PBS at 37 °C for 30 min, wells were washed once with PBS. Subsequently, wells were coated with 100 µL of 500 ng/mL ICAM-1-Fc in PBS at 37 °C for 1 h. Afterwards, wells were gently washed twice with PBS. Mock- or HSV-infected mDCs were harvested 24 hpi, washed once with RPMI 1640 and a final cell concentration of 4 × 10^5^ cells/mL in RPMI 1640 was adjusted. Per well, 4 × 10^4^ mDCs/100 µL were subjected to the adhesion assay. After an incubation of 45 min at 37 °C, the non-adherent fraction was gently washed off with pre-warmed RPMI 1640. Each experimental condition was performed in quadruplicates. Input cells and adherent fractions were quantified using the β-glucuronidase assay as described above (“Fibronectin adhesion assay”).

### 2.11. Transwell Migration Assay

Transwell inserts, with a membrane pore size of 5 µm (Corning Costar, Bodenheim, Germany), were coated with 20 µg/mL fibronectin at 4 °C overnight. The next day, the fibronectin solution was discarded and the transwell inserts were equilibrated using 100 µL DC-medium. To the lower wells of a 24-well plate, 600 µL DC-medium were added. Transwell migration assays were conducted as previously described [57,88]. Briefly, mock- or HSV-/HSV UV-infected mDCs were harvested 4 or 24 hpi, washed once in PBS and adjusted to a cell concentration of 2 × 10^6^ cells/mL in DC-medium. To eliminate cell clusters, cell suspensions were passed three times through a 20 gauge needle using a syringe. For the migration assay, 2 × 10^5^ mDCs/100 µL were seeded onto the inserts and cells were allowed to migrate at 37 °C for 2 h toward the chemokine CCL19 (final concentration 100 ng/mL; Peprotech), added to the lower well. Additionally, 100 µL of the cell suspension was preserved as input condition. Afterwards, the migrated fractions were harvested from the lower well. Input samples and the respective migrated fractions were quantified using the β-glucuronidase assay as described above (“Fibronectin adhesion assay”).

### 2.12. Immunofluorescence Confocal Microscopy

After 16 h of infection, mock- or HSV-2-infected mDCs, with or without MG-132 treatment, were allowed to adhere on poly-L-lysin (Sigma Aldrich) coated glass cover slips at 37 °C for 1 h. Subsequently, cells were fixed with 4% PFA in PBS. Afterwards, cells were gently washed with PBS and permeabilized using 0.2% Triton-X-100 in PBS for 10 min. Blocking of unspecific antibody binding was performed using 1% BSA in PBS for 20 min. Antibodies used for immunofluorescence staining: primary rat hybridoma anti-CYTIP clone 2F9 or rat hybridoma anti–cytohesin-1 clone 7H2 and secondary anti-rat AlexaFluor555-conjugated antibody (Invitrogen, Carlsbad, USA), and primary mouse anti-ICP0 (clone 11060, Santa Cruz Biotechnology) and secondary anti-mouse AlexaFluor488-conjugated antibody (Invitrogen, Carlsbad, USA). For mounting and nuclear staining of the cells, Roti^®^-Mount FluorCare DAPI (CarlRoth) was used. Samples were analyzed by confocal microscopy using a LSM780 microscope (Zeiss). 

### 2.13. Statistical Analyses

For determination of the significance, multiple data sets were analyzed using one-way analysis of variance (ANOVA). Results are displayed as means ± standard deviations (SD) or as standard error of the mean (SEM) as indicated. Significance was accepted for *p*-values less than 0.05. **** indicates *p* ≤ 0.0001; *** *p* ≤ 0.001; ** *p* ≤ 0.01; * *p* ≤ 0.05; and ns, not significant.

### 2.14. Approvals and Legal Requirements

A positive vote from the local ethics committee has been obtained (reference number 184_16) for the generation of monocyte-derived DCs generated from leukapheresis products (LRSCs) of healthy donors. This study was carried out in accordance with the recommendations of the ethics committee of the “Friedrich-Alexander-Universität Erlangen-Nürnberg” with written informed consent from all subjects including the accordance with the Declaration of Helsinki.

## 3. Results

### 3.1. HSV-2 Successfully Establishes Viral Protein Expression in iDCs and mDCs

For in vitro generation of DCs, peripheral blood mononuclear cells obtained by leukapheresis from healthy donors were purified and monocytes were differentiated into DCs by cultivation in IL-4- and GM-CSF-conditioned medium. The resulting iDCs were either infected with HSV-2 strain G or HSV-1 as control, and harvested at 2, 4, 8, or 16 hpi or matured into mDCs, via stimulation with a maturation cocktail, additionally containing IL-1β, IL-6, TNF-α, and PGE2, followed by the same infection procedure as for iDCs. To investigate whether HSV-2 is able to efficiently infect mDCs, Western blot analyses were performed for detection of proteins specific for each of the three viral gene expression phases, i.e., immediate early (ICP0, ICP4, ICP27), early (ICP8), or late (gB; Figure 1).

As expected, infection of iDCs or mDCs with HSV-1 revealed strong expression of all analyzed viral proteins in a time-dependent manner. HSV-2-infected iDCs and mDCs showed high expression levels of the herpesviral proteins tested. However, the expression onset of the probed viral proteins seemed to be slightly delayed in HSV-2- compared to HSV 1-infected DCs until 4 hpi. After 16 h, DCs either infected with HSV-1 or HSV-2 displayed equal amounts of viral protein expression of all three phases. Thus, we concluded that monocyte-derived iDCs and mDCs were efficiently infected with HSV-1 and HSV-2 in vitro.

### 3.2. HSV-2 Inhibits DC Maturation When Infecting iDCs

In a subsequent step, we aimed to analyze whether HSV-2 inhibits the maturation process of iDCs, similar to HSV-1, under the herein applied experimental conditions. Therefore, iDCs were generated and either infected with HSV-2 strain G, HSV-1 as control, or mock-treated. One hour post infection cells were cultivated in medium supplemented with a defined maturation cocktail to induce DC maturation. DCs were analyzed via flow cytometry for the expression of distinct surface markers at 24 hpi. As controls, iDCs prior to infection as well as mDCs two days after induction of maturation were included (Figure 2, Appendix A).

Compared to their immature counterparts the expression levels of the analyzed surface molecules, except for the general DC marker CD11c, were strongly induced on 24 h (mock) as well as 48 h matured DCs. In sharp contrast, and in line with previous reports [89,90], both HSV-1 and HSV-2 significantly inhibit this increase in the expression levels of distinct maturation-associated proteins on DCs. Based on these results, we conclude that our protocol for the differentiation and maturation of DCs facilitates the in vitro generation of iDCs and mDCs, which can be successfully infected and analyzed during an HSV-1 or HSV-2 infection. 

### 3.3. HSV-2 Induces the Proteasome-Dependent Degradation of CD83 in mDCs 

Next, we tested for protein expression of the well-described maturation marker CD83 on HSV-2-infected mDCs. This glycoprotein was previously shown to undergo strong proteasomal degradation upon HSV-1 and HCMV infection of mDCs [79,91]. To examine whether HSV-2 similarly triggers the downmodulation of CD83 on/in mDCs, we performed HSV-2 infection studies with or without application of the proteasomal inhibitor MG-132. Subsequently, flow cytometric analyses of surface-expressed CD83, CD86, or CD11c, and Western blot experiments for the verification of total CD83 levels were conducted (Figure 3 and Appendix A).

First, flow cytometric analyses revealed that not only HSV-1 but also HSV-2 mediates the downregulation of CD83 surface expression, which can be blocked by inhibition of the proteasome using MG-132 (Figure 3A). To exclude a general decrease of surface proteins, we additionally determined CD86 (Figure 3B) and CD11c (Figure 3C) surface expression. While CD86 levels were unaffected on both HSV-1- and HSV-2-infected mDCs compared to the mock control, CD11c expression was significantly increased upon an HSV-1 but not HSV-2 infection 16 hpi. 

Second, we also observed a strong downregulation of CD83 in protein lysates of HSV-2-infected mDCs, which was rescued by treating the cells with MG-132. These combined data indicate that also HSV-2 mediates the proteasome-dependent degradation of the functionally important CD83 molecule in mDCs, very similar to HSV-1.

### 3.4. HSV-2 Inhibits CCL19-Directed Transwell Migration of mDCs Early upon Infection

Previously, our group demonstrated that HSV-1 inhibits two- as well as three-dimensional migration of infected mDCs toward the chemokine CCL19 [57,88]. This chemokine is highly expressed in secondary lymphoid organs (SLOs) to guide migrating cells via their chemokine receptor CCR7 (in)to these tissues. Thus, inhibition of migration toward these areas with high lymphocyte density enables HSV-1 to prevent an immune response and therefore support establishment of latency. To study whether this observation is unique for HSV-1 or also found for other α-herpesviruses, HSV-2 was analyzed regarding its impact on DC adhesion and migration. 

We performed transwell migration assays as a model system to determine the migration capability of HSV-2-infected versus HSV-1-infected and mock-treated mDCs toward the CCL19-dependent chemotactic signal. Spontaneous migration without the presence of any chemokine served as control, indicated as “w/o”. After two hours, migrated cells were harvested in the lower wells and quantified by measuring their β-glucuronidase activity. Results are shown as percentages of migrated mDCs relative to the respective input condition (Figure 4).

Figure 4 clearly indicates successful induction of transwell migration of mock-infected cells (white bars) upon addition of CCL19, in comparison to spontaneous migration without any chemokine, at 24 hpi (Figure 4A) as well as 4 hpi (Figure 4B). In line with our recent findings, HSV-1 (black bars) drastically inhibited spontaneous and CCL19-mediated mDC chemotaxis 24 hpi (Figure 4A) [57,88]. Residual migration capability of infected mDCs was less than 10% of input cells, compared to approximately 90% of mock-treated cells (white bars). More importantly, the data of this experiment revealed HSV-2 to equally hamper CCL19-directed mDC migration under the herein used conditions compared to HSV 1 at 24 hpi.

To further study the HSV-mediated inhibition of mDC migration, transwell migration assays were performed at early time points post infection (Figure 4B), since HSV causes multiple changes regarding DC biology 24 hpi, e.g., chemokine receptor downmodulation [88] or induction of apoptosis [92]. Thus, mDCs were mock-, HSV-2 or HSV-2 UV-infected and harvested 4 hpi. Cells were subsequently subjected to transwell migration assays toward CCL19, equally conducted as described for Figure 4A. Treatment of mDCs with UV-inactivated HSV-2 virions at an MOI of 5 slightly reduced mDC transwell migration capability toward CCL19 to 70%, while infection with HSV-2 strongly inhibited mDC chemotaxis to a residual migration capability of only 10%. 

Taken together, these data demonstrate that also HSV-2 significantly inhibits mDC transwell migration toward CCL19, with fast kinetics from 4 hpi onwards. Moreover, also spontaneous migration of HSV-2-infected mDCs was equally inhibited (Figure 5, “w/o”), which rather suggests a chemokine-independent mechanism of hampering mDC transwell migration at this very early time point post infection.

### 3.5. HSV-2 Mediates the Reduction of CCR7 Surface Expression Late during mDC Infection

Pathogen-mediated modulation of chemokine receptor surface expression constitutes a mechanism to hijack cell migration, since proper expression of chemokine receptors is a prerequisite for directed chemotaxis toward the respective chemokine. Previously, our group demonstrated that HSV-1 significantly reduces the expression of the CCL19-sensing chemokine receptor CCR7 on mDCs from 8 hpi onwards [88]. Hence, expression levels of the chemokine receptor CCR7 was monitored on mock-, HSV-1- and HSV-2-infected mDCs via flow cytometry at 4 and 24 hpi to verify whether the observed block in CCL19-directed migration at 4 hpi is associated with reduced CCR7 levels or not (Figure 5 and Appendix A). Additionally, we stained for CD83 serving as a known downmodulated target during an HSV-1 or HSV-2 infection of mDCs.

The CD83 surface expression followed a distinct pattern on HSV-1- versus HSV-2-infected mDCs, becoming significant on HSV-1-infected mDCs from 4 hpi onwards, while being time-delayed on HSV-2-infected mDCs, compared to mock controls (Figure 5). In contrast, while the surface expression of CCR7 was unaffected on HSV-1- and HSV-2-infected mDCs, compared to their uninfected counterparts at 4 hpi, we observed a clear reduction at 24 hpi. At this late time point post infection, HSV-1- and HSV-2-infected mDCs showed approximately 50% and 20% of residual surface expression of CCR7 receptor expression, respectively.

Based on the unaffected CCR7 surface expression at 4 hpi we hypothesized that distinct mechanism(s), apart from solely downregulating the respective chemokine receptor CCR7, account for the HSV-mediated inhibition of mDC transwell migration at this early time point post infection. In the following, one potent HSV-mediated mechanism will be discussed, which interferes with mDC migration very early upon infection. However, later during infection, HSV-1 and HSV-2 also hamper CCR7 expression as an additional counterstrike for delaying mDC migration toward the SLO-expressed chemokine CCL19.

### 3.6. HSV-2-Infected mDCs Show Increased Adhesion

As previously described for HSV-1-infected mDCs, inhibition of mDC migration not only depends on proper expression of specific chemokine receptors but also on the precise regulation of mDC adhesion [57]. It is therefore reasonable to assume that also in the context of an HSV-2 infection, increased adhesion might contribute to the inhibition of mDC migration. Thus, mock-treated and HSV-1-, HSV-1 UV-, HSV-2-, or HSV-2 UV-infected mDCs were harvested 4 hpi and subjected to adhesion assays on the extracellular matrix protein fibronectin. As a positive control, mock-infected cells were incubated with Mg/EGTA to induce the activity of the β2 integrin LFA-1 and thus adhesion (Figure 6A). Treatment with Mg^2+^ ions promotes a conformational change from the inactive into the active epitope, while simultaneous presence of the chelator EGTA specifically depletes Ca^2+^ ions, which inhibit β2 integrin activity [87,93]. Moreover, the treatment with Mg/EGTA selectively induces the activated epitope of the β2 integrin LFA-1, but not that of Mac1 [87]. 

In an alternative approach, we subjected mock-treated and HSV-1- or HSV-2-infected mDCs to adhesion assays on the LFA-1 ligand ICAM-1 at 24 hpi (Figure 6B). Adherent fractions were quantified by assessing their β-glucuronidase activity, after allowing the cells to adhere for 45 min. The quantity of adherent cells was calculated based on the respective input samples and relative to mock treated cells (set to “1”).

The treatment of mDCs with Mg/EGTA strongly induced their adhesion on fibronectin coated wells up to approximately 3.5-fold compared to untreated mock controls. An even higher induction of mDC adhesion was observed for HSV-1- (4.5-fold) and also HSV-2-infected mDCs (five-fold), whereas mDC adhesion was unaffected upon incubation with the same amount of UV-inactivated virions at 4 hpi (Figure 6A). These data demonstrate that HSV-2—equally to HSV-1—increases fibronectin adhesion of infected mDCs very early after infection. Furthermore, HSV-2 also induced ICAM-1 adhesion of mDCs up to three-fold compared to uninfected cells. Since ICAM-1 is a specific ligand for β2 integrins, these combined data indicate that infection of mDCs with HSV-2 fosters β2 integrin-dependent adhesion.

### 3.7. Surface Expression of β2 Integrin Subunits is Differentially Regulated on HSV-2-Infected mDCs

Members of β2 integrins are abundantly expressed on mDCs and important mediators of fibronectin and ICAM-1 adhesion. Especially LFA-1, composed of CD11a and CD18, was reported to be influenced during an HSV-1 infection of mDCs [57]. To study whether an HSV-2 infection modulates the expression levels of β2 integrin subunits on mDCs, flow cytometric analyses were performed. The respective subunits CD11a/b/c and CD18 were analyzed on mock-, HSV-1-, HSV-1 UV-, HSV-2-, and HSV-2 UV-infected mDCs at an early (4 hpi) as well as a late (24 hpi) time point post infection. Values are shown as percentages relative to the mock control (Figure 7 and Appendix A).

After four hours of infection, CD11a/b/c surface expression levels were unaffected in any of the indicated infection conditions. In contrast, CD18 surface expression was approximately 1.5-fold upregulated on HSV-1-but not on HSV-2-infected mDCs (Figure 7D). Later during infection, i.e., 24 hpi, surface expression levels of CD11c revealed a weak induction on HSV-1-, HSV-2-, and HSV-2 UV-infected mDCs compared to mock cells. Moreover, HSV-1- and HSV-2-infected mDCs displayed an approximately three-fold upregulation of CD18.

These data clearly show that the surface expression of β2 integrin subunits, especially CD11a and CD18 comprising LFA-1, were unaffected on HSV-2-infected compared to mock-treated mDCs 4 hpi. Hence, another mechanism apart from solely increasing β2 integrin subunit surface expression accounts for increased mDC adhesion at this very early time point post infection.

### 3.8. HSV-2-Infected mDCs Display Increased β2 Integrin Activity

Considering that integrins are constitutively expressed but not constitutively active, their activation status might be modulated during an HSV-2 infection of mDCs, which was previously described in the context of HSV-1 [57]. Thus, the activation status of β2 integrins on HSV-2-infected mDCs was analyzed 4 hpi as well as 24 hpi (Figure 8 and Appendix A). As controls, HSV-1-infected as well as Mg/EGTA-treated mock controls were further included in this experiment. To investigate the ligand binding state of β2 integrins, the antibody mAB24 that exclusively recognizes the activated epitope was used for flow cytometric analyses [94,95,96].

As expected, both control conditions, i.e., HSV-1-infected as well as mock Mg/EGTA-treated mDCs, revealed a significant induction of β2 integrin activity of approximately two- to three-fold. In agreement with this, also HSV-2-infected mDCs displayed strong β2 integrin induction, compared to mock cells, already 4 hpi. This activation further increased during the onset of infection. In contrast, incubation of mDCs with either HSV-1 or HSV-2 UV-inactivated virions did not affect β2 integrin activity 4 hpi, whereas the HSV-2 UV-condition showed a significant induction at 24 hpi. 

In conclusion, HSV-2 induces β2 integrin activity on infected mDCs similarly to HSV-1 very early after infection. This observation correlates with the onset of HSV-2-mediated induction of mDC adhesion and reduction of mDC transwell migration.

### 3.9. HSV-2 Mediates Rapid Downmodulation of CYTIP Protein Expression

Given the importance of cytohesin-1 and CYTIP in the regulation of β2 integrin activity we analyzed their expression levels during the course of an HSV-2 infection. Therefore, mDCs were mock- or HSV-2-infected, or HSV-1-infected as a control, and harvested 4, 8, 16, and 24 hpi for subsequent Western blotting (Figure 9).

In accordance with previous reports, HSV-1-infected mDCs showed progressive reduction of CYTIP protein expression from 4 hpi onwards, compared to their uninfected counterparts (Figure 9A) [57]. Interestingly, also HSV-2 negatively influenced CYTIP protein levels. CYTIP downmodulation was detected from 4 hpi onwards, with an almost complete absence of CYTIP protein expression at 24 hpi. Moreover, by probing the blots with the αCYTIP hybridoma supernatant, we detected an additional band of approximately 55 kDa that increases upon infection and might constitute post-translationally modified CYTIP protein. Apart from this, cytohesin-1 protein levels were only slightly affected upon an HSV-1 or HSV-2 infection of mDCs. Furthermore, while treatment with HSV-1 UV-inactivated virions did not severely hamper the expression of CYTIP or cytohesin-1, HSV-2 UV-inactivated virions mediated a notable loss of their expression levels from 16 hpi onwards.

Combining these data, it is tempting to speculate that the HSV-2-mediated reduction of mDC migration and the induction of mDC adhesion rapidly upon infection are due to CYTIP downmodulation from 4 hpi onwards.

### 3.10. HSV-2 Induces a Proteasome and Ubiquitin-Dependent Degradation of CYTIP

To mechanistically analyze the HSV-2-mediated downmodulation of CYTIP protein levels, we tested whether inhibition of the proteasome restores CYTIP protein levels in infected mDCs. Therefore, mock- or HSV-2-infected mDCs, or HSV-1 as control, were treated with or without 10 µM of the proteasomal inhibitor MG-132 at 1 hpi. Cells were harvested 16 hpi and subsequently lysed for Western blot analyses to detect CYTIP as well as cytohesin-1 protein expression (Figure 10A).

Infection of DMSO-treated mDCs with either HSV-1 or HSV-2 resulted in a marked reduction of CYTIP protein levels compared to the respective mock condition. Conversely, cytohesin-1 expression was only marginally reduced in HSV-1- as well as HSV-2-infected mDCs. Interestingly, loss of CYTIP expression was significantly restored in MG-132-treated infected mDCs. In order to further verify these data, we performed immunofluorescence staining for CYTIP or cytohesin-1 in mock-treated versus HSV-2-infected mDCs with or without addition of MG-132 16 hpi (Figure 10B). Consistent with our Western blot results, protein levels of CYTIP were restored upon presence of MG-132 during an HSV-2 infection, whereas cytohesin-1 protein levels were almost unaffected in any of the indicated conditions. Combining these data, HSV-2 triggers the proteasomal degradation of CYTIP in infected mDCs similar to its family member HSV-1 [57].

Finally, we analyzed whether the proteasomal degradation of CYTIP is dependent on ubiquitination. For this, HEK293T cells were transfected with plasmids coding for CYTIP or cytohesin-1, since this cell type showed stable expression of CYTIP upon treatment with PYR-41, an inhibitor of the ubiquitin E1-activating enzyme. After 24 h cells were either mock-treated or infected with HSV-2 or HSV-2 UV-inactivated virions, or HSV 1 as a control. One hour post infection, 10 µM MG-132 or 2 µM bortezomib (BZ) were added to the medium to inhibit the proteasome. For inhibition of ubiquitination 80 µM PYR-41 were added 4 hpi. Cells were harvested 18 hpi for Western blot analyses of CYTIP (left panels) and cytohesin-1 (right panels) expression levels (Figure 11).

Consistent with our data obtained from mDC infection experiments, we observed a strong decrease in CYTIP protein expression in HSV-1- as well as HSV-2-infected HEK293T cells in the absence of any inhibitor, whereas cytohesin-1 protein levels remained stably expressed (Figure 11). Moreover, treatment of HEK293T cells with UV-inactivated HSV-1 or HSV-2 virions did not alter the expression levels of CYTIP or cytohesin-1. Notably, inhibition of proteasomal degradation using MG-132 or bortezomib restored CYTIP protein expression upon HSV-1 or HSV-2 infection also in HEK293T cells. More importantly, inhibition of the ubiquitination cascade using PYR-41 blocked HSV-1- and HSV-2-mediated degradation of CYTIP in transfected HEK293T cells.

In summary, HSV-2 not only blocks maturation of DCs when infecting iDCs, but also triggers the proteasome-dependent degradation of CD83 as well as CYTIP in mDCs. The latter observation very likely accounts for the induction of adhesion and inhibition of migration capability of infected mDCs rapidly upon infection. Together, these modulations of HSV-2-infected DCs are reminiscent of those mediated by HSV-1 and reflect the immunomodulatory capacity of HSV to hamper the induction of potent antiviral immune responses.

## 4. Discussion

During millions of years of co-existence and co-evolution, not only the host was compelled to develop a multitude of defense mechanisms, but also pathogens—such as viruses—evolved strategies to subvert or hamper those host immune responses [97,98]. HSV infections induce significant changes in the host’s proteome, some of them targeting immune-related pathways [99]. Despite the probably most important strategy of immune evasion, i.e., establishment of latency and lifelong persistence [62], HSV has evolved additional strategies to escape the clearance by the host’s antiviral immune responses [100]. During primary HSV infection, the virus productively replicates in initially infected epithelial cells, e.g., keratinocytes [62,101], but also immune cells such as iDCs are encountered by the virus. Since the release of distinct “danger” signals, from directly infected cells, can trigger the maturation of adjacent uninfected iDCs [102,103,104], mDCs constitute one of the subsequent target cells that can also be infected by HSV. During their maturation, DCs switch from an antigen-sampling iDC phenotype into a migratory antigen-presenting mDC phenotype, while the latter one is able to induce an adaptive immune response. Thus, pathogens and especially persistent viruses hamper crucial functions of DCs to delay the initiation of an antiviral immune response [92,105].

In previous publications it was suggested that HSV-1 might not be able to complete its gene expression profile in mDCs [106,107]. This hypothesis was due to the observation that progeny virus was barely detectable in cell culture supernatants of HSV-1-infected mDCs, in contrast to those of iDCs [83,106]. However, we have previously shown that HSV-1 completes its gene expression profile both in iDCs and mDCs, since infected mDCs exhibited viral protein expression of the immediate-early, early, and late phase (Figure 1) [108]. In accordance, also HSV-2 efficiently infects iDCs and mDCs exhibiting similar kinetics of viral protein expression (Figure 1) [90]. Upon infection, we and others showed that HSV-1 and HSV-2 hamper crucial functions of DC biology. First, infection of iDCs with either of both herpesviruses inhibits their maturation into antigen-presenting migrating mDCs (Figure 2) [89,90]. This renders infected DCs unable to present HSV-derived antigens or to migrate toward draining lymph nodes for subsequent T cell activation. Interestingly, also HCMV is capable of inhibiting iDC maturation, which therefore represents an immune evasion mechanisms shared by different Herpesviridae members [77]. Second, when infecting mDCs, HSV-2—equal to HSV-1—mediates a strong decrease in surface as well as intracellular expression levels of CD83, a protein implicated in the modulation of T cell responses (Figure 3) [11,91,106,109]. Briefly, HSV-1 mediates the proteasome-dependent degradation of CD83 in mDCs [91], which was also proven for HSV-2 in the present study (Figure 3). However, whether the HSV-2-mediated CD83 degradation also follows an ICP0-triggered and ubiquitin-independent pathway, as shown for HSV-1 [91,110], remains to be elucidated. Interestingly, targeting surface-expressed CD83 for its downmodulation is also shared by the α-herpesvirus varicella-zoster virus (VZV) and β-herpesvirus HCMV [78,79,111]. Third, infection of mDCs with HSV-1 and HSV-2 also leads to a downregulation of surface expression of the chemokine receptor CCR7 late during infection (Figure 5) [88]. Since the signaling axis among the chemokine receptor CCR7 and its cognate chemokine CCL19 is pivotal for directed mDC migration toward T cell zones in lymph nodes [16,17], its downmodulation constitutes an important countermeasure of mDC migration capability [88]. Consistently, despite the presence of directly HSV-infected DCs at the site of initial infection and an overall increased migration of DCs to draining lymph nodes in vivo [112,113], infected DCs are barely detectable in the migrating population [114,115]. The latter observation indicates that HSV-infected DCs are indeed impaired in their migratory capacity in vivo. In this context, lymph-node resident and submucosal DCs were found to be crucial for antigen-specific T cell priming via cross-presentation of antigens from migratory cells [112,116,117,118,119].

Thus, our in vitro experiments help to better understand why directly HSV-infected DCs are absent from the migrating population in vivo [114,115]. Previously, we have shown that not only the inhibition of CCR7 surface expression but also the rapid induction of mDC adhesion triggers the inhibition of mDC in vitro migration very early upon an HSV-1 infection [57]. Interestingly, here we show that CCL19-dircted transwell migration, using fibronectin-coated inserts, is also significantly inhibited when mDCs were infected with HSV-2, compared to the respective mock condition (Figure 4). Moreover, also spontaneous migration, i.e., without any chemotactic signal, was drastically reduced. This indicates that a mechanism independent of chemokine (receptor) signaling is additionally involved. This hypothesis is further strengthened by the observation that decreased migration capability occurred already 4 hpi, while the chemokine receptor CCR7, mediating CCL19-directed migration, was downregulated at later time points post infection (Figure 4 and Figure 5) [88]. In conclusion, HSV-2—similarly to HSV-1—significantly reduces migration of mDCs in vitro. This is not only due to reduced chemokine receptor signaling, but also to an additional HSV-mediated modulation of mDCs. Importantly, an inhibition of CCL19-directed mDC transwell migration was recently reported also in the context of a β-herpesvirus infection, i.e., HCMV [80]. In sharp contrast, and very interestingly, another member of α-herpesviruses, VZV, does not interfere with mDC migration. This facilitates successful spread and dissemination of VZV inside the host for subsequent infection of T cells, accompanied by severe modulations of both DCs and T cells [120].

Since proper mDC migration requires loosening of strong adhesive forces characteristic of their immature counterparts, adhesion has to be tightly controlled during the life cycle of DCs. Theodoridis et al. (2011) reported that the surplus of adhesive contacts caused the impaired migration capability of HSV-1-infected mDCs in migration experiments in vitro [57]. Here we reveal that also HSV-2 induces mDC adhesion to fibronectin (Figure 6A), which can be bound by β1 as well as β2 integrins [39,121], and to ICAM-1 (Figure 6B), serving as specific ligand for β2 integrins [36,40]. This HSV-2-mediated induction of mDC fibronectin adhesion was higher compared to the treatment of mock cells with Mg/EGTA, which selectively induces the activity of the β2 integrin LFA-1 [87]. It is tempting to speculate that, despite an involvement of further β2 or β1 integrins, LFA-1-dependent adhesion predominantly accounts for the observed increase in mDC adhesion during an HSV-2 infection, as demonstrated for HSV-1 [57]. It is well known that β2 integrins constitute the most abundantly expressed integrin subfamily on leukocytes and therefore possess a dominant role in controlling mDC adhesion [36,122]. Since any modulation of their expression levels or activity can affect DC adhesiveness, and in turn migration, their state of activation has to be tightly regulated [34]. Previously, Theodoridis and co-workers reported that mDC adhesion is significantly influenced by an altered β2 integrin activity, in particular LFA-1, upon an HSV-1 infection [57]. In accordance with this, the here presented data reveal that, despite unaffected expression levels of the β2 integrin subunits on HSV-2-infected until 4 hpi (Figure 7), the active conformation of β2 integrins was strongly induced (Figure 8). Due to the fact that integrins are ubiquitously expressed but not constitutively active, their ligand binding affinity is bi-directionally regulated via inside-out and outside-in signaling events [46,51,123]. Obviously, HSV-2 hijacks this tight regulation of β2 integrin activity in order to promote mDC adhesion. The finding that the onset of HSV-2-induced β2 integrin activation timely correlates with increased adhesion, and in turn reduced transwell migration, strengthens this hypothesis. Supporting the observation that β2 integrins are the major mediators of increased mDC adhesion upon HSV infection, a previous publication pointed out that DCs are more dependent on β2 integrin activation compared to other leukocytes [124].

Activation of β2 integrins via inside-out signaling comprises distinct pathways, which include the intracellular binding of specific proteins to the β2 integrin subunit CD18. In leukocytes, particularly DCs, two proteins directly promote the conformational change of β2 integrins into the activated epitope—namely talin and cytohesin-1 [46,124,125,126]. While talin is able to bind to the β subunit of several integrin subfamilies [127], cytohesin-1 is a specific β2 integrin binding partner [124]. Cytohesin 1 binding to the CD18 subunit leads to a conformational change into the high affinity state, promoting integrin activation, ligand binding, and cell adhesion [53,54,55,124]. In steady state, the direct interaction partner of cytohesin-1, CYTIP, regulates the intracellular localization of cytohesin-1. Upon specific triggers, such as chemokine signaling, CYTIP is recruited to the plasma membrane, where it interacts with cytohesin-1 to abrogate the cytohesin-1-mediated activation of β2 integrins via cytoplasmic translocation of the cytohesin-1/CYTIP complex [56,125]. Previously, it has been demonstrated that siRNA-mediated ablation of CYTIP in mDCs causes a significant induction of fibronectin adhesion and inhibited migration capacity, indicating an inverse correlation between CYTIP expression and the strength of adhesion as well as migration [57,58]. Thus, it was tempting to speculate that HSV might regulate the expression levels of at least one of these proteins to increase β2 integrin activity, i.e., upregulation of cytohesin-1 versus downregulation of CYTIP. Interestingly, the γ-herpesvirus Kaposi’s sarcoma-associated herpesvirus (KSHV) encodes a cytohesin-1 homologue, kaposin A, to support cytohesin-1 membrane localization and therefore CD18 binding [128]. Since HSV-2 lacks a kaposin A-like protein, a distinct mechanism had to account for the observed elevated levels of activated β2 integrins (Figure 8). Consistent with HSV-1-infected mDCs [57], HSV-2-infected mDCs reveal only moderate regulation of cytohesin-1 protein expression later during infection (Figure 9). However, CYTIP protein expression was significantly downregulated upon an HSV-2 infection, which followed a proteasome-and ubiquitin-dependent mechanism (Figure 9, Figure 10 and Figure 11). Notably, the here presented data are consistent with previous reports analyzing CYTIP degradation in HSV-1- or HCMV-infected mDCs [57,80]. Importantly, the onset of CYTIP downmodulation in HSV-2-infected mDCs at 4 hpi timely correlated with the observed increase in β2 integrin activity as well as adhesion and reduction of migration (Figure 4, Figure 6, Figure 8, Figure 9 and Figure 10).

## 5. Conclusions

The here presented data of HSV-2-infected mDCs extended and strengthened recent reports regarding CYTIP degradation, induction of adhesion, and reduction of migration of HSV-1- and HCMV-infected mDCs [57,80]. Based on these reports, hampering antigen delivery to SLOs and thus inhibiting the induction of a potent antiviral immune response seems to be evolutionary conserved among distinct members of the Herpesviridae. Future studies are required to elucidate the molecular mechanism of HSV-1-, HSV-2-, and HCMV-mediated proteasomal degradation of CYTIP in mDCs, e.g., involved viral and cellular proteins. This might help to develop new or improve existing antiviral drugs/therapies, to target the HSV-/HCMV-mediated inhibition of mDC migration. However, the development of such antiviral therapies is impeded by a plethora of additional immune evasion mechanisms evolved by HSV and HCMV [77,78,92,105,106,129].

## Figures and Tables

**Figure 1 viruses-12-00112-f001:**
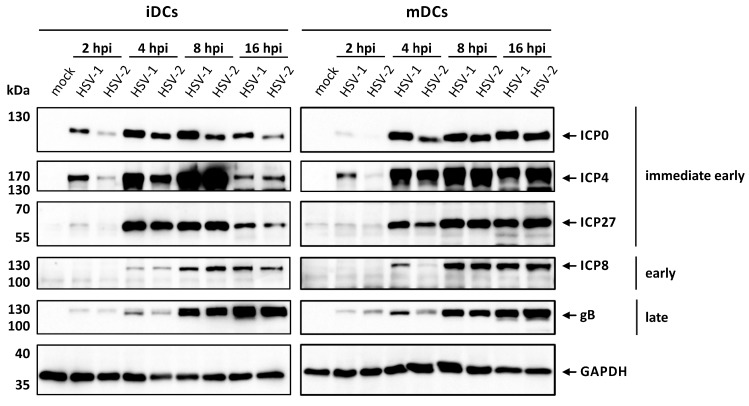
HSV-2 initiates viral protein expression in immature and mature dendritic cells (DCs). Immature or mature DCs were mock-, HSV-1- (multiplicity of infection (MOI) of 1), or HSV-2-infected (MOI of 5) and harvested at the indicated time points post infection. Protein lysates were prepared and equal amounts were loaded on a 12% acrylamide SDS-gel, which was subjected to Western blotting. Protein levels of HSV-1/2 immediate early proteins ICP0, ICP4, and ICP27, early protein ICP8 as well as late protein gB were detected. GAPDH was used as loading control. Representative data out of two experiments are shown.

**Figure 2 viruses-12-00112-f002:**
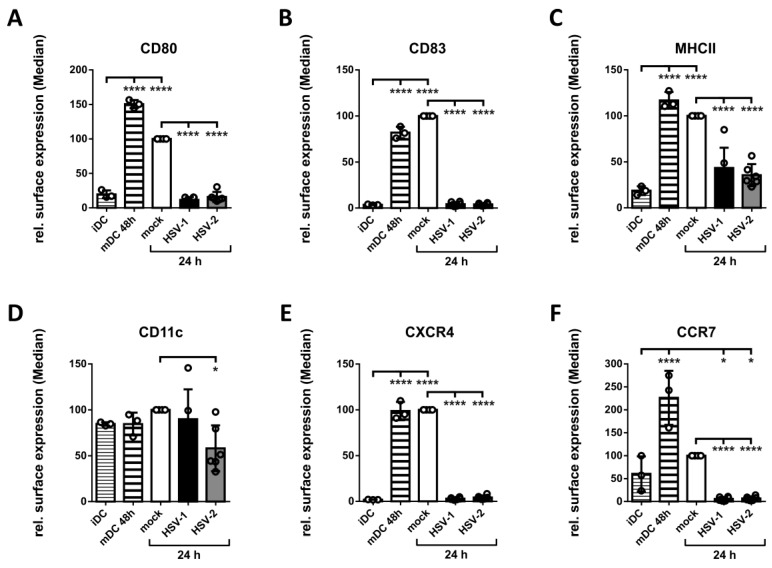
HSV-2 inhibits DC maturation. Immature DCs were generated and either directly used for flow cytometric analyses, serving as input control, or mock-, HSV-1- (MOI of 2) and HSV-2-infected (MOI of 5) followed by addition of the maturation cytokine cocktail. After 24 h, DCs were subjected to flow cytometric analyses. Additionally, 48 h matured mock-treated DCs served as a positive control. The cells were stained with specific fluorochrome-labeled antibodies directed against (**A**) CD80, (**B**) CD83, (**C**) MHCII, (**D**) CD11c, (**E**) CXCR4, and (**F**) CCR7. Median values are shown as percentages relative to 24 h matured mock-treated DCs. The experiment was performed at least three times with cells from different healthy donors. Error bars indicate ± SD. Significant changes (**** = *p* < 0.0001; * = *p* < 0.05) are marked by asterisks.

**Figure 3 viruses-12-00112-f003:**
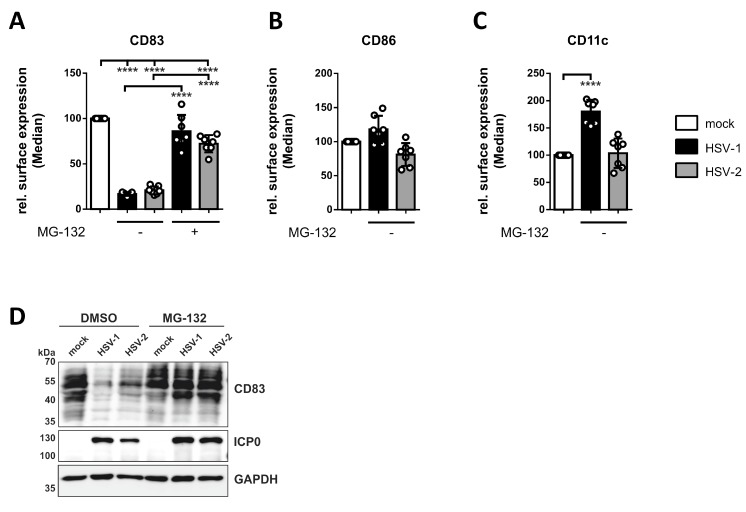
Loss of CD83 expression on/in HSV-2-infected mature DCs (mDCs). Mature DCs were mock-, HSV-1- (MOI of 2), or HSV-2-infected (MOI of 5) and treated with 10 µM MG-132 at 1 hpi, or DMSO as control. DCs were harvested 16 hpi and (**A**–**C**) subjected to flow cytometric or (**D**) Western blot analyses. (**A**–**C**) The cells were stained with specific fluorochrome-labeled antibodies directed against (**A**) CD83, (**B**) CD86, and (**C**) CD11c. Median values are shown as percentages relative to mock-treated DCs. The experiment was performed seven times with cells from different healthy donors. Error bars indicate ± SD. Significant changes (**** = *p* < 0.0001) are marked by asterisks. (**D**) Protein lysates were prepared and equal amounts were loaded on a 12% acrylamide SDS-gel, which was subjected to Western blotting. Protein levels of CD83, HSV-1/2 immediate early protein ICP0, and GAPDH were detected. Representative data out of four experiments are shown.

**Figure 4 viruses-12-00112-f004:**
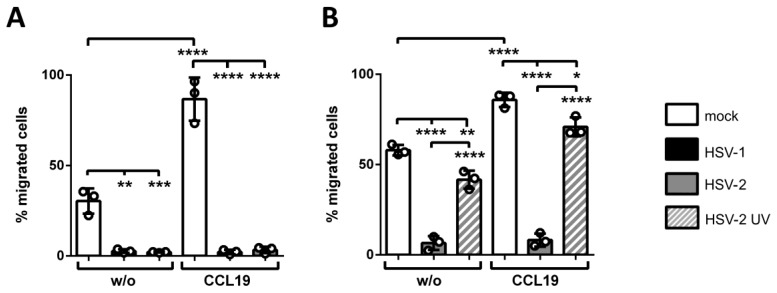
HSV-2 inhibits mDC transwell migration toward a CCL19 chemokine-gradient early upon infection. (**A**) Mature DCs were mock-, HSV-1- (MOI of 2) or HSV-2-infected (MOI of 5) and harvested 24 hpi. (**B**) Mature DCs were mock-, HSV-2- (MOI of 5), or HSV-2 UV-infected (MOI of 5; 1.200 J/cm^2^) and harvested 4 hpi. (**A**,**B**) Cells were subjected to transwell migration assays on fibronectin-coated transwell inserts toward a CCL19 chemokine-gradient (100 ng/mL). Spontaneous migration without addition of chemokines (“w/o”) in the lower wells served as control. Migration capacity was determined after 2 h and percentages of migrated cells were quantified by assessing their β-glucuronidase activity. Mock- (white columns), HSV-1- (black columns), HSV-2 (gray columns), and HSV-2 UV-infected (gray striped columns) mDCs are depicted. The experiments were performed three times with cells from different healthy donors. Error bars indicate ± SD. Significant changes (**** = *p* < 0.0001; *** = *p* < 0.001; ** = *p* < 0.01; * = *p* < 0.1) are marked by asterisks.

**Figure 5 viruses-12-00112-f005:**
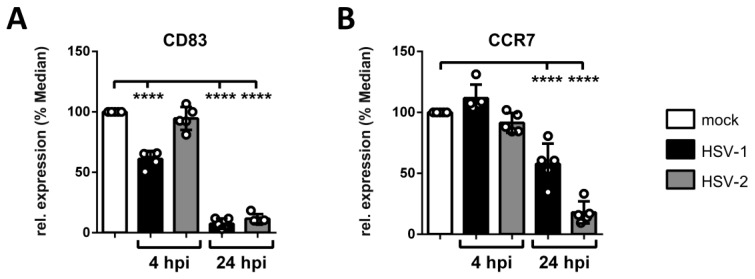
HSV-2 reduces the surface expression of the chemokine receptor CCR7 on mDCs 24 hpi. Mock-, HSV-1- (MOI of 2), or HSV-2-infected (MOI of 5) mDCs were harvested 4 hpi and 24 hpi. Cells were stained with antibodies specific for (**A**) CD83 or (**B**) CCR7 for flow cytometric analyses. Median values of HSV-1-(GFP-positive; black bars) and HSV-2-infected cells (gray bars) are depicted relative to mock-treated cells (white bars; set to 100%). The experiment was performed five times with cells from different healthy donors. Error bars indicate ± SD. Significant changes (**** = *p* < 0.0001) are marked by asterisks.

**Figure 6 viruses-12-00112-f006:**
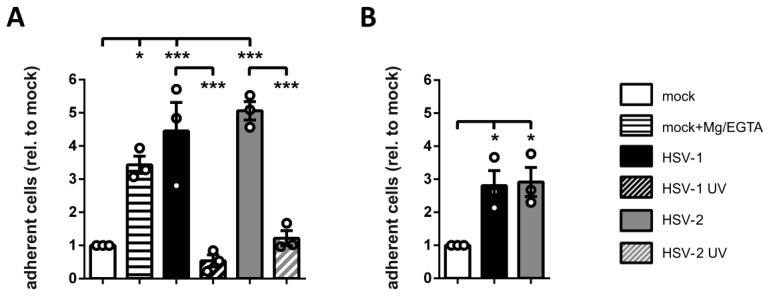
HSV-2 induces fibronectin and ICAM-1 adhesion of mDCs. (**A**) Mock- (white bars), HSV-1- (MOI of 2, black filled bars), HSV-1 UV- (MOI of 2; 8 × 0.12 J/cm^2^, black striped bar), HSV-2- (MOI of 5; gray filled bars), or HSV-2 UV-infected (MOI of 5; 8 × 0.12 J/cm^2^, gray striped bar) mDCs were harvested 4 hpi. Mock controls were treated with or without Mg/EGTA. Cells were allowed to adhere on fibronectin-coated wells for 45 min. (**B**) Mature DCs were mock- (white column), HSV-1- (MOI of 2, black column), or HSV-2-infected (MOI of 5, gray column) and harvested 24 hpi. Cells were allowed to adhere to ICAM-1-Fc coated plates for 45 min. (**A**,**B**) Input conditions as well as adherent cells were quantified by measuring the β-glucuronidase activity. Changes in mDC adherence are shown relative to the mock condition (set to “1”).The experiment was performed three times with cells from different healthy donors, while each single condition was performed in quadruplicates. Error bars indicate ± SEM. Significant changes are marked by asterisks (*** = *p* < 0.001; **p* < 0.05).

**Figure 7 viruses-12-00112-f007:**
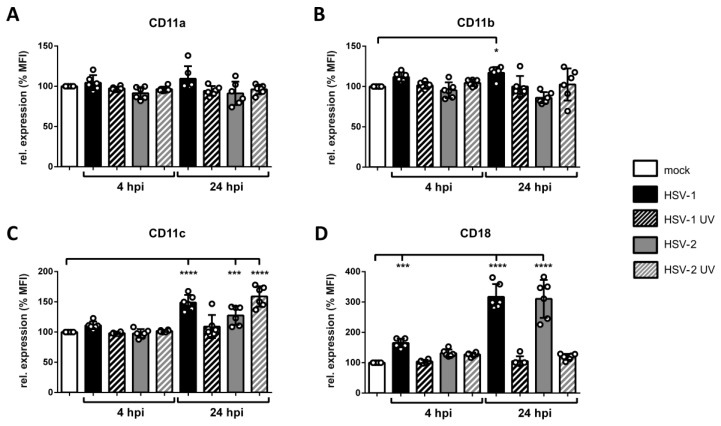
Expression levels of β2 integrin subunits are differentially regulated on HSV-2 infected mDCs 4 hpi and 24 hpi. Mock-, HSV-1- (MOI of 1), HSV-1 UV- (MOI of 1; 1.200 J/cm^2^), HSV-2- (MOI of 5), or HSV-2 UV- (MOI of 5; 1.200 J/cm^2^) infected mDCs were harvested 4 hpi and 24 hpi. Cells were stained with antibodies specific for (**A**) CD11a, (**B**) CD11b, (**C**) CD11c, or (**D**) CD18 and analyzed via flow cytometry. Panels show data as mean fluorescence intensity (MFI) for HSV-1- (black filled bars), HSV-1 UV- (black striped bars), HSV-2 (gray filled bars), or HSV-2 UV- (gray striped bars) infected mDCs relative to mock cells (white bars; set to 100%). Error bars indicate ± SD. Significant changes (**** = *p* < 0.0001; *** = *p* < 0.001; * = *p* < 0.05) are marked by asterisks. The experiment was performed six times with cells from different healthy donors.

**Figure 8 viruses-12-00112-f008:**
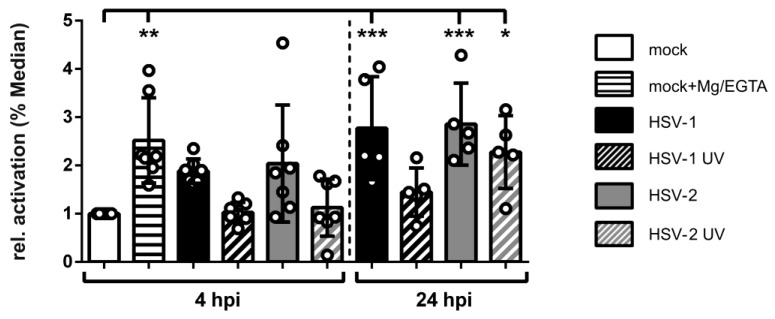
HSV-2 induces β2 integrin activity very early after infection. Mock- (white bars), HSV-1- (MOI of 2, black filled bars), HSV-1 UV- (MOI of 2; 8 × 0.12 J/cm^2^, black striped bar), HSV-2- (MOI of 5; gray filled bars), or HSV-2 UV-infected (MOI of 5; 8 × 0.12 J/cm^2^, gray striped bar) mDCs were harvested 4 hpi and 24 hpi. As positive control, mock-treated cells were treated with or without Mg/EGTA 4 hpi as indicated. Cells were stained with the antibody mAb24 specific for the activated epitope of CD11a/CD18 (β2) integrins, or the respective isotype control, and analyzed via flow cytometry. Changes in β2 integrin activity for the indicated conditions are shown as ΔMedian (mAb24-isotype) relative to mock-infected mDCs (white bar; set to “1”). The experiment was performed five (24 hpi) to seven (4 hpi) times with cells of different healthy donors. Error bars indicate ± SD. Significant changes (*** = *p* < 0.001; ** = *p* < 0.01; * = *p* < 0.05) are marked by asterisks.

**Figure 9 viruses-12-00112-f009:**
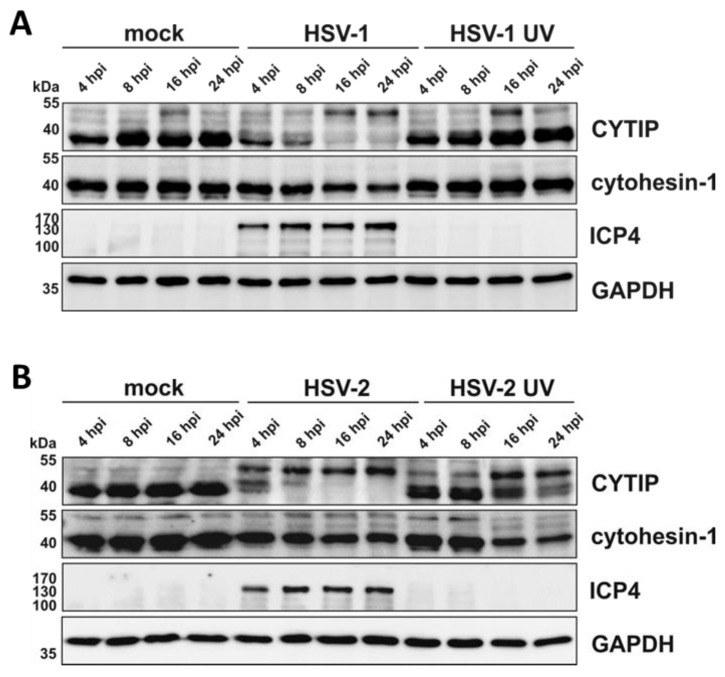
Downmodulation of CYTIP protein levels in HSV-1- and HSV-2-infected mDCs occurs rapidly upon infection. Mature DCs were mock-, (**A**) HSV-1- (MOI of 2), HSV-1 UV- (MOI of 2; 8 × 0.12 J/cm^2^), (**B**) HSV-2- (MOI of 5), or HSV-2 UV-infected (MOI of 5; 8 × 0.12 J/cm^2^) and harvested at the indicated time points post infection. Protein lysates were prepared and equal amounts were loaded on a 12% acrylamide SDS-gel, which was subjected to Western blotting. Protein expression of CYTIP, HSV-1 ICP0/4 as infection control and GAPDH as loading control were detected. The experiment was performed three times with cells from different healthy donors and representative data are shown.

**Figure 10 viruses-12-00112-f010:**
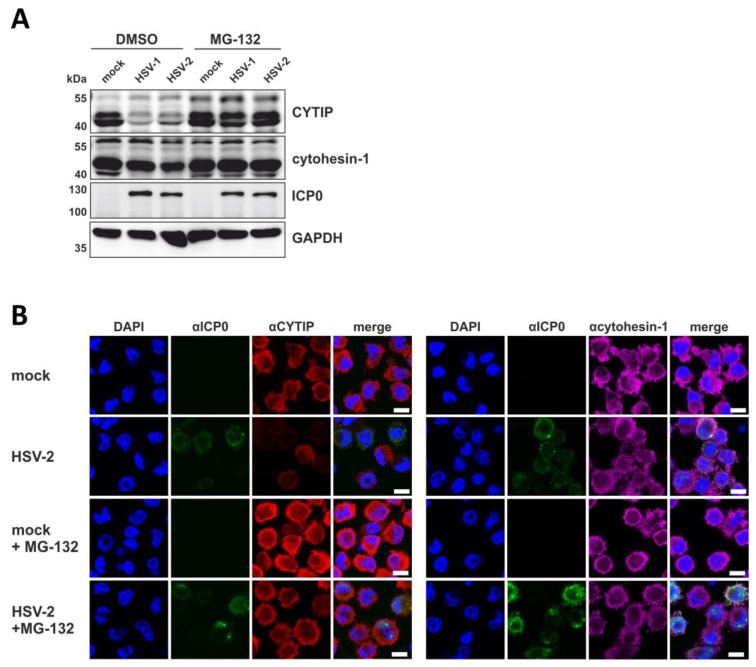
HSV-2 induces proteasomal degradation of CYTIP in mDCs. Mock-, HSV-1- (MOI of 1), or HSV-2- (MOI of 5) infected mDCs were treated with or without the proteasomal inhibitor MG-132 (10 µM) and harvested 16 hpi. As a control, cells were treated with DMSO. (**A**) Protein lysates were subjected to Western blot analysis using equal protein amounts loaded on a 12% acrylamide SDS-gel. Expression levels of CYTIP, cytohesin-1, ICP0 as infection control, or GAPDH as loading control, were monitored using specific antibodies. (**B**) Immunofluorescence staining using an αCYTIP (left panel) or αcytohesin-1 (right panel) primary antibody followed by staining with an AlexaFluor555-tagged secondary antibody was performed. Infection was visualized by using a primary antibody specific for HSV ICP0 followed by staining with an AlexaFluor488-coupled secondary antibody. The nucleus was stained using Dapi. The scale bar indicates 10 µm. The experiments were performed three times with cells from different healthy donors and representative data are shown.

**Figure 11 viruses-12-00112-f011:**
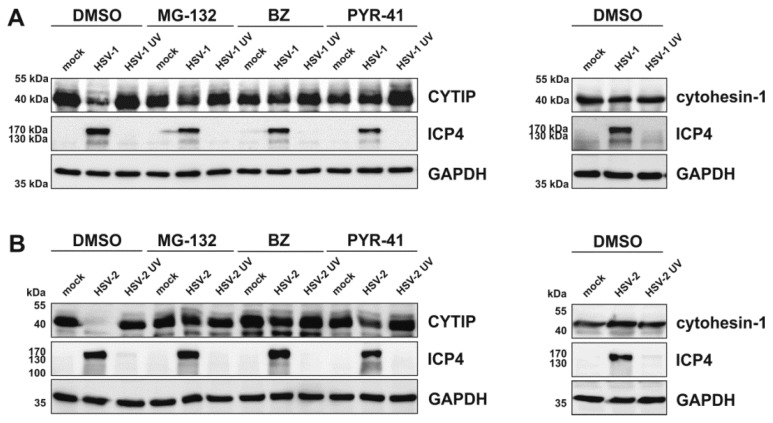
HSV-1 and HSV-2 induce proteasome- and ubiquitin-dependent degradation of CYTIP in transfected HEK293T cells. HEK293T cells were transfected with 1 µg of plasmid DNA encoding CYTIP (left panels) or cytohesin-1 (right panels). After 24 h, cells were mock-treated or infected with (**A**) HSV-1 (MOI of 2) or HSV-1 UV-inactivated virions (MOI of 2; 8 × 0.12 J/cm^2^) and (**B**) HSV-2 (MOI of 5) or HSV-2 UV-inactivated virions (MOI of 5; 8 × 0.12 J/cm^2^). Cells were treated with or without the proteasomal inhibitors MG-132 (10 µM) or bortezomib (BZ; 2 µM) 1 hpi, or the ubiquitination inhibitor PYR-41 (80 µM) 4 hpi. As control, cells were treated with DMSO. Cells were harvested 18 hpi and protein lysates were subjected to Western blot analysis using equal amounts of protein loaded on a 12% acrylamide SDS-gel. Expression levels of CYTIP, cytohesin-1, ICP4 as infection control, or GAPDH as loading control, were monitored using specific antibodies. The experiment was performed three times independently and representative data are shown.

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
