# Peer review of "Herpes Simplex Virus Type-2 Paralyzes the Function of Monocyte-Derived Dendritic Cells"

_viruses, 2020, doi:10.3390/v12010112_

Round 1
Reviewer 1 Report
Dear Authors
The article by Grosche et al., systematically analyses the effect of HSV infection with monocyte derived DC model and shows the functional impairment of dendritic cells. The work is well performed with suitable experimental models and approaches. I think it will be great to give some clarifications and that will enable the readers to have better understanding about the work.
Most of the FACS data is represented as MFI and it will be great to show the real FACS data ( as supplementary fig) to demonstrate the actual number of DCs infected by the HSVs. Do you have 100 % infection in all the experiments? If you are working with a mixed population of DCs, (for eg:- 60 % not infected vs 40 % infected and that can make a difference in the observed results). The above comment is applicable for all other FACS MFI data and it will be great if the readers can see the real expression of each molecule on the DCs under different treatment conditions. I kindly request you to add the FACS plots showing the percentage expression of each makers as a supplementary data. This will give an idea to the readers how homogenous is your population and that will make your results even better. Does the infections is equivalent in each experimental conditions with the Mock vs VSVG. If you carefully look at the figure 2 mock –DCs have comparable maturation in 24 hrs time period with control DCs activated for 48 hrs. What exactly the mock? Do you have any data shows the mock vs uninfected DCs have a similar level of surface phenotype after activation. Does the mock itself makes the DCs a better DCs due to activation and increase the significance of the difference and the HSV infected DCs exhibit a profile similar to iDCs. So if you can show the FACS data of infection with mock and HSV, as well as including the migration data and other similar data of non-infected DCs or iDC as a control will be great. What is the CD83 or other expression level of mock transduced and not activated DCs? Most of the data is presented shows a comparable functional impairment on DC infected with HSV 1 or 2. Authors always stress in titles that the HSV2 role in functional impairment. Authors did all the statistical comparison with the mock control and not with HSV 1 vs HSV2. The main difference observed is on the CCR7 expression. The authors should describe it as a HSV and not specifically HSV2 or both HSV1 and 2. Authors use dendritic cells in the title and it will be nice if mentioned as Monocyte derived DCs. There clear experimental results shows some of the DC subsets are not susceptible for viral infections (PMID: 28783704) and the HSV are also susceptible for the RAB15 mediated mechanism. The work clearly demonstrate that the HSV infection induce a protein degradation and what makes CD18 to overcome the degradation
Thanks
Reviewer 2 Report
In their article „Herpes simplex virus type-2 paralyzes dendritic cell function“, Grosche et al. investigate the impact of HSV-2 infection on dendritic cell (DC) characteristics and function. Similar results have been shown for DC infected with HSV-1. The findings are clearly presented, however, a few points in the manuscript should be clarified.
P. 7, Fig. 1. The authors show that various viral proteins are expressed upon DC infection. Why is an ICP27 band visible in the mock-infected sample? While the authors state that viral proteins are detected in a time-dependent manner, gB, a late protein, is already detected at 2 hpi. What is the explanation for this early detection? Further, it is stated in the discussion that few viral particles can be found in supernatants of HSV-1 infected cells. Did the authors also measure HSV-2 particles in supernatants?
P. 9, Fig. 3. CD80 expression levels are unaffected by HSV-2 without MG-132 (p. 9, ll. 375/376), however, they are affected as shown in Fig. 2A although data were analyzed at different time points (16 hpi vs. 24 hpi). Was the sample in Fig. 3 analyzed in the presence or absence of MG-132?
P. 10, Fig. 4. How do the authors explain the high percentage of migrated mock-infected DC?
P. 11, Fig. 5 and ll. 454ff. A block in CCL-19-directed migration at 4 hpi occurs despite regular CCR7 surface expression. What „distinct mechanisms“ do the authors have in mind?
P. 14, ll. 556/557. Please check the sentence.
Figure numbers on p. 16 and p. 17 should be 10 and 11, respectively.
P. 19, l. 711 should read „characteristic of“.
The colours chosen for the HSV-1 and HSV-2 bars in the figures are hard to distinguish and should be changed.
